# “You Do It without Their Knowledge.” Assessing Knowledge and Perception of *Stealthing* among College Students

**DOI:** 10.3390/ijerph17103527

**Published:** 2020-05-18

**Authors:** Marwa Ahmad, Benjamin Becerra, Dyanna Hernandez, Paulchris Okpala, Amber Olney, Monideepa Becerra

**Affiliations:** 1Department of Health Science & Human Ecology, California State University, San Bernardino, CA 92407, USA; moham300@coyote.csusb.edu (M.A.); 005150037@coyote.csusb.edu (D.H.); pokpala@csusb.edu (P.O.); aolney@csusb.edu (A.O.); 2Department of Information & Decision Sciences, California State University, San Bernardino, CA 92407, USA; bbecerra@csusb.edu; 3Center for Health Equity, California State University, San Bernardino, CA 92407, USA

**Keywords:** sexual consent, college students, condom, privacy, sexually transmitted diseases, infections

## Abstract

In recent years, the act of nonconsensual condom removal, termed stealthing, has become commonly discussed on social and print media; yet, little to no evidence exists on the current knowledge and perception of stealthing among young adults. As such, we assessed what college students know and feel regarding stealthing. We employed an exploratory mixed-method analysis where focus groups were followed by a quantitative survey. A qualitative assessment was conducted using grounded theory analyses and questions for a quantitative survey were developed based on emergent themes from focus groups. Quantitative data was analyzed using descriptive and bivariate analyses with alpha less than 0.05 to denote significance. Though limited knowledge exists, participants felt that stealthing was a violation of their privacy, trust, sexual consent, and their ability to make a health decision, and should be considered an assault. Participants noted stealthing may have become acceptable due to its popularity in social media and young adult culture, especially porn. We also found sex differences in the perception of stealthing being considered a sexual assault with lower rates among males as compared to females. Our results demonstrate that there is a need for health educators to assess the prevalence of such a behavior among college students and policy makers to assess the legal implications of nonconsensual condom removal.

## 1. Introduction

Sexual consent is commonly defined as freely agreeing to participate in sexual activity; though the legal definition varies by state in the U.S. [1] For example, in California, consent is defined as free and voluntary agreement with knowledge of the act, while North Carolina has no specific definition of consent, although threat to bodily harm is considered lack of consent. Likewise, while California requires the definition of consent to include words such as “freely given” or “affirmative,” similar requirements are not present in Texas or Arizona, making legal and consistent understanding of consent difficult across the nation [1]. In recent years, the importance of sexual consent has gained further interest—often attributable to the #metoo movement—that has brought to light the importance of discussing consent and violence related to sexual health [2].

Despite national attention on sexual consent, a novel term that has arisen, especially through social media, has been that of stealthing, currently defined as the act of nonconsensual condom removal [3,4]. Stealthing has often been referred to as gift giving or generationing, and is associated with men who have sex with men (MSM) where one partner who is HIV-positive infects the other without the latter’s knowledge [5]. While the practice of stealthing is not new, the terminology has been promoted through the internet beyond that of MSM or HIV, especially through social media and has recently been reported as a rising trend, as further noted in empirical literature [3]. Specifically, corners of the internet have discussed the practice and stealthing has progressively become a trending new topic. In fact, various posts on social media promote men bragging about their act of nonconsensual condom removal. According to social media platforms such as reddit, men brag about having the right to “spread their seeds” [6]. Further, Brodsky’s legal analysis [4] found that men discuss the subject of stealthing and support one another with tips and tricks of doing such acts or encourage other men to stealth their partners. While the term stealthing was not used, the act of sexual intercourse without a condom, which contradicted the agreement of a sexual act between two adults, came to international light in 2011 under the legal case Assange v Swedish Prosecution Authority (2011) EWHC 2849. In this case, Assange was accused of rape due to having sexual intercourse without a condom, and while previously the court ruled against that conditional consent, in order to ensure extradition, the court ruled against Assange, thus making conditional consent legal under English law [7].

While peer-reviewed studies in the U.S. are lacking, a study on stealthing prevalence in Australia noted that 32% of women and 19% of MSM who attended health service clinics reported experiencing stealthing [8]. In the U.S., a study addressed the term stealthing, though it was utilized in the context of non-consensual HIV transmission among MSM. Though not directly related to the more novel and rising social media use of the word stealthing, the results of the study do note that 4.6% of the sample experienced similar behavior [5]. Likewise, among 503 women aged 21–30 years, Davis et al. found that 19% of the participants experienced coercive condom use resistance (CUR). Though this does not directly relate to the stealthing behavior promoted on social media, it does shed light on the common practice of sexual acts without condoms when consent is lacking [9].

To date, however, not only is there no state or federal law in the U.S. explicitly addressing stealthing, there is also limited empirical research on stealthing, knowledge and attitudes towards it, its prevalence, or its impact, especially among young adults. Moreover, a recent article by Ebrahim [3] noted that the practice of nonconsensual condom removal is intertwined with “sexual autonomy, sexual consent, and sexual violence,” and warrants further academic research, though there remains a significant paucity of literature. In this study, we thus provide empirical evidence of young adults’ knowledge and perception of stealthing and in turn produce a foundation for evidence-based policy implications and further research on the driving factors of the behavior.

Our study was driven by the theoretical foundations of the growing field of sexual health literacy [10,11], which goes well beyond general health literacy and how patients navigate the healthcare system. While often utilized among sexual minorities, sexual health literacy addresses questions of having accurate knowledge related to sexual health, how one manages their own sexual health and wellbeing, as well as structural and contextual influences that may impact sexual knowledge and health.

## 2. Methods

We undertook an exploratory sequential mixed-methods [12] approach where qualitative data was first analyzed followed by quantitative data to further elucidate key themes that emerged from the former. Particularly, survey questions were developed on content that were deemed to be of interest for further evaluation based on focus group results of emergent themes. For example, the concept of sexual consent was consistent and predominant in the focus groups as an emergent theme, thus survey questions were added to assess how participants felt about giving and asking for consent and their related self-efficacy (i.e., confidence in asking and giving consent). Reporting of all qualitative results followed the standard thematic analysis supported by quotes, but quantitative data is presented in applicable numeric numbers and *p* value when appropriate.

### 2.1. Sample and Data Collection

College students from a medium-sized four-year public university were selected as the study population. General education courses through multiple departments gave out extra credit as an incentive to participate. Per Institutional Review Board approval guidelines, any registered student aged 18 years or older was allowed to participate in the study. No student was prevented from participating in the study and all students received extra credit deemed non-coercive by the Institutional Review Board.

Data was collected in two phases: focus groups and survey questionnaire. All participants agreeing to attend focus groups were first given consent forms to sign and for those giving consent, data collection continued. Participants were asked semi-structured interview questions for a total of one hour. Interviews were led by two researchers with training in qualitative assessment. Focus group questions were centralized on understanding the knowledge of stealthing and sexual consent, as well as influences of stealthing behavior. Focus group interviews were digitally recorded and transcribed verbatim. Each focus group was limited to no more than five participants. Central questions for the focus groups were based on the following questions, though it should be noted that the formatting of some questions was slightly modified or probe questions were added based on participant feedback:What is stealthing?Do you think stealthing increases one’s risk of STI (sexually transmitted infection) transmission and how?Do you think stealthing is considered a form of sexual assault and why?Do you think stealthing should be against the law and why?What are your thoughts on stealthing being socially acceptable?

For the second phase using a survey questionnaire, the participants were distinct from those in the qualitative analyses to ensure a lack of diffusion of questions and thus protect against bias results. Participants were given consent forms and had the right to refuse to participate in the study without consequences to their grade in the course or any other academic outcomes. The survey was anonymous and a separate sign-in sheet ensured extra credit was given, as approved by the Institutional Review Board. Surveys were given out in general education courses and included both closed and open-ended questions assessing knowledge, attitude, and self-efficacy related to sexual consent, as these were emergent themes from the qualitative analyses.

### 2.2. Data Analysis

We analyzed transcripts from focus groups and results of open-ended questions from surveys using a modified grounded theory. Grounded theory has been modified over the years from traditional, to evolved, to constructivist [13]. While traditional grounded theory’s goal was to generate a conceptual theory that accounts for patterns of behavior, the purpose of our study was to assess knowledge and perception. Thus, a modified version was more appropriate where constructivist grounded theory addressed how participants construct meaning towards a particular topic (in our case, stealthing).

Each member of the research team independently read the transcripts and created a list of commonly occurring phrases and words as preliminary codes. This method was repeated until no new codes were generated. Such codes were then reviewed for patterns and themes, followed by labeling and the naming of domains. As per grounded theory protocol, we continued this analysis until no new themes emerged, thus reaching saturation, and focus group recruitment was concluded.

Quantitative data from survey questionnaires were input into SPSS software v24 (IBM Corp.; Armonk, NY, USA) for analysis. Descriptive statistics were conducted to assess the prevalence of variables of interest, including sexual consent-related knowledge, attitude, and self-efficacy. Sex differences were further assessed using bivariate analyses (chi-square test for independence), with alpha less than 0.05 to denote significance. Missing values were excluded from analyses.

## 3. Results

A total of 13 participants were included in focus groups before theoretical saturation was reached. A total of 153 students participated in the quantitative survey. Qualitative assessment of the focus group and open-ended survey questions highlighted several emergent domains (Table 1). These included: knowledge themes of awareness and how a lack of it can impact health-decision making; consent with multiple themes of communication and violation/assault; and finally, social norms and acceptance of stealthing.

### 3.1. Knowledge

A majority of the participants were unable to define the word stealthing, but once the term was explained many of them associated it with a negative connotation, specifically the impact of health-decision making. Participants in our study consistently noted that condoms are essentially used to protect an individual from STIs and HIV; when individuals agree to have consensual sex with a condom, they are essentially protecting themselves from unwanted diseases. The moment that a condom is removed, both parties are no longer protected, which may impact their sexual health. Although the act of stealthing is not new, the terminology of the act is. Participants were aware of the act but did not know that was the term used to describe a condom removal without the consent of the partner. Among the 153 participants that were surveyed, a small percentage knew the act of stealthing, but a majority associated a negative connotation; however, there were no significant sex differences among individuals who participated.
“…I have no idea, never heard of it. It sounds kind of negative in a way”.

Once participants had the meaning of stealthing explained to them, a lack of knowledge continued to be a theme, especially around its impact on health-decision making.
“Not everybody knows what they have and not everybody gets tested so they won’t know what they are passing on to other people, you could get gonorrhea, syphilis, or HIV.”

In addition, some participants were aware of the act, but not necessarily the terminology itself.
“…You do it without their knowledge.”
“Oh that’s what it’s called?”

Among the 153 participants in the survey, 5.2% reported that stealthing was the removal of a condom without consent, while 8.5% associated the word with a negative connotation, with no significant sex differences (the rest identified it incorrectly) to the open ended question: *What does the word stealthing mean?*

### 3.2. Consent

The lack of consent was another emerging domain, with further themes of trust, communication, privacy, and violation. While the interview questions in focus groups did not ask about consent, participants consistently brought up consent; which is aligned with current literature highlighting the relation of stealthing to the concept of lack of consent [3]. Many participants expressed that removing a condom without the knowledge or consent of the partner was going against the sexual act that the person agreed on. The terms of agreement of the sexual act changed once the condom is removed without the knowledge or consent of the partner, which led to a violation of trust and a demonstrates poor communication between both individuals. Communication between two individuals is very important; an individual should be told or asked if they want the condom to be removed during a sexual activity, that way they have the choice whether they want to continue or not. Not giving them the opportunity to choose violates what they intentionally consented to do and puts the individual at a risk of contracting an STI or HIV.
“…basically again they took off the condom without your consent and then continued to do something that you did not agree to do, the terms changed.”
“I don’t think it’s right because you both agreed that you were going to do it with a condom, but it’s not right for either one of them to take off the condom and continue with the sexual activity without the person knowing there is no longer a condom.”

Furthermore, participants highlighted that such break of consent and trust further put the health of those impacted at risk.
“It’s a violation of their consent, they should have been told if they wanted to take off the condom that way they have the choice whether they want to continue or not. It is a violation of their trust, privacy, and it’s a health concern as well.”

A third emergent theme under the domain of consent was the consideration of stealthing as sexual assault. Participants expressed that stealthing should be considered a violation or sexual assault, as once the condom is removed without a partner’s consent, the act of stealthing is similar to rape.
“The second the page is split and you both are on different pages there is an issue, as soon as you step over that boundary and remove the condom without your partner’s knowledge that is considered sexual assault.”
“I feel once either partners says no or stop or I don’t want this anymore the moment the condom is removed without their knowledge that’s when it becomes sexual assault.”
“I feel like it is assault because if something were to happen like a pregnancy or sexual transmitted disease there is repercussions and consequences and someone has to answer to that.”
“Yes, because maybe that one person has HIV and they give it to you, then you are stuck with it. I mean people don’t think of it as serious, but if that person is having sex with a lot of people and they do that [stealthing], then they are putting you at a higher risk of getting HIV and if you end up getting it then someone is to blame.”

In the quantitative assessment (Table 2), participants were asked if unwanted touching, kissing, or hugs should be considered sexual assault. While a majority reported strongly agree/agree, a lower percent was found among males as compared to females (89.5% vs. 96.4%, *p* < 0.05). Similar differences were further noted in response to the question regarding whether condom removal without the approval of a sexual partner should be considered sexual assault. While a majority reported strongly agree/agree, a lower percent was noted for males compared to females (82.1% vs. 85.4%, *p* < 0.05).

### 3.3. Social Norm

Another emergent theme was the perception that if stealthing is perceived to be a social norm, then the practice is more likely to be prevalent. For example, some participants expressed confusion because in porn, which can be socially normal for young adults, there is no way to affirm if consent was asked or not prior to the removal of a condom. Participants consistently stated that since a lack of a condom is prevalent in porn, it can encourage curiosity, especially if it is normalized in porn.
“I’ve watched porn and I think it’s weird that some don’t use condoms since that’s their career and they do that a lot, but one time I saw that in a video that a guy was wearing a condom, but then he took it off, but I don’t know if he asked the girl for consent or if she was forced.”
“The first time someone sees stealthing in a porn video they might think it’s a normal thing to do and if they see it so often it can encourage them to probably try it too.”
“…monkey see monkey do.”

Participants expressed that stealthing in the porn industry has the same correlation as that of video games encouraging violence in children.
“…not to stray off with video games, but it can apply to stealthing in porn as well. People see video games and say “Oh my kid is violent now”, you can imply the same exact thing with porn, “oh you watch porn this way, and so you’re going to have sex this way.”

Participants noted that individuals who watch porn do not realize that the act of stealthing is wrong because of a lack of awareness or a lack of education regarding risk factors that is associated with the behavior.
“If they don’t know it’s wrong then they are not going to see it as wrong. They are going to see it as, this is something fetishized and something that I might like so I am going to try it. But if there is educational explanation that stealthing shouldn’t be fetishized and the risk factors and consequences that are associated to stealthing, they would less likely try it.”

### 3.4. Self-Efficacy

A majority of participants reported strongly agree/agree when asked about their ability to ask and give consent during sexual activity, as well as their ability to say no during sexual activity (Table 3). No statistical difference was noted by sex.

## 4. Discussion

The results of our study highlight several key emergent domains. One major emergent domain was the lack of knowledge about the term stealthing, even while students were familiar with the behavior itself. Such results on low knowledge of various sexual health-related factors is similar to that in the literature discussing other sexual health issues, including HIV, sexual consent, etc. For example, Moore and Smith noted that while college students tend to be very knowledgeable on HIV, they are less knowledgeable regarding STIs and general sexual health, as well as on preventive measures [14]. In particular, King et al. noted that while overall STI knowledge was low among college students belonging to a minority group, length of stay in the United States was related to higher knowledge, thus indicating that minority immigrant college students may have better STI knowledge upon exposure to such education [15]. Thus, given that our student population was primarily an immigrant group, targeted health education initiatives to address sexual health literacy—especially condom use—may be imperative to ensure that the most at-risk populations have appropriate resources.

The second emergent domain was on consent. When asked about their confidence (self-efficacy) in asking or giving consent during sexual activity, more than half of participants felt confident. However, the lack of actual understanding what is included in consent is evident from the aforementioned assessment of unwanted touching, kissing, etc. For example, while a majority felt confident giving and receiving consent, they did not identify unwanted kissing and touching as a part of consent; which contradicts the United Nations’ inclusion of such acts as sexual harassment—[16], a term that is broader than assault [17]—which was commonly cited by our participants. We also noted that a higher percentage reported confidence in asking or giving consent than being able to say no to sexual activity. As noted across various platforms [17,18,19], saying no at any time during sexual activity is a part of sexual consent. Thus, while participants feel confident asking and giving consent, when asked about a key component of consent, such as saying no, we see differences. This demonstrates a disconnect between understanding a general term versus specific components of it. A similar trend is noted when dealing with addressing whether unwanted behavior should be considered assault. As mentioned earlier, despite high rates of participants reporting confidence in giving and asking consent, a lower percentage felt that unwanted behavior should be classed as sexual assault, further demonstrating a disconnect from understanding a general concept versus specific components of it. Given that sexual harassment is an umbrella term that includes sexual abuse, assault, etc., and given that the United Nations includes such unwanted behavior as part of sexual harassment, the results noted in our study highlight the need for further sexual health education. Cumulatively, these results show that college students may feel that they understand the general concept of consent, but may lack the knowledge of what is actually included in consent such that sexual harassment can be avoided. As such, health education efforts for college students would benefit from discussing state-specific consent definitions as well as what entails sexual harassment. While we did note some sex differences in the quantitative study which was not apparent in the qualitative study, it is feasible that the open discussion in mixed-sex focus groups was a factor, and thus future studies could benefit from sex-specific focus groups.

The third domain was that of social norms where the participants reported that watching porn, which is common among college students, often normalized the lack of a condom and stealthing was common in such porn videos. The literature notes that normalization of a negative behavior can further increase such a behavior among young adults. For example, Neighbors and groups found that college students who overestimate the drinking of their peers and perceive their friends as more approving of alcohol tended to practice a similar behavior [20]. Likewise, normalization of alcohol use through social media has been further shown to be related to adolescents’ positive attitudes towards alcohol [21]. This is consistent with what we see among our participants, as many expressed that if stealthing is perceived as a social norm, then the practice is more likely to be prevalent, especially among individuals who may lack knowledge or education of risk factors associated with the act of stealthing. As such, the literature and the results of our study further note the importance of addressing stealthing in the social context. Moreover, anecdotal evidence notes that a Google search of the key words of “porn” and “stealthing” will lead to a plethora of videos and sites including it, though formalized research is needed. Regardless, the importance of the connection between porn and stealthing was highlighted in Australia’s bill [22] where stealthing was included as part of banning and criminalizing revenge porn. While the authors of this manuscript acknowledge that participants’ perceptions of porn as an influence on stealthing remains to be supported by literature, it does warrant further research to potentially interview stealthers (i.e., those that engage in the act of stealthing) and their driving factors.

This further brings to attention significant legal cases occurring globally that are related to addressing the legal implications of stealthing, which has been further highlighted in the literature as an area of debate [3,23]. While the purpose of our study was to assess the knowledge and perception of stealthing and not legal implications, results of focus group interviews have given rise to participants’ discussion of legal consequences. As such, this warrants some discussion, though further legal research is needed. For example, in Switzerland, a male who met a female through a dating platform was convicted for deliberating removing a condom without his partner’s knowledge. While they both agreed to have sex, the female refused unprotected sex and later discovered that the condom was removed without her consent during sexual activity [24]. Likewise, in Germany, a German police officer was found guilty of sexual assault for removing a condom without the consent of his partner and was sentenced to eight months in jail, fined $3400, and had to pay for a sexual health test for the female victim [25]. Such cases highlight two major themes: there is little understanding what stealthing is in the legal context and there remains little precedence for the legal consequences. As noted in our study population, sexual assault was a consistent emergent theme in relation to the legal consequences of stealthing, though participants may also have considered rape as an option for legal charges. However, participants defined sexual assault differently, with some including rape, threats, and unwanted sexual contact, while others (mostly females) considered unwanted touching, kissing, or hugging as sexual assault as well. This further affirms that not only is there a need for a comprehensive and concrete definition of consent, but also that of sexual assault, that in turn can pave the way for more legal precedence in such cases. Whether stealthing should be criminalized has been met with debate and further comprehensive legal research is needed before such measures are made. For example, Blanco [23] notes that while the removal of a condom without consent may negate the bodily autonomy of women, considering it as rape may instead be over-criminalization. As such, a major recommendation has been to avoid new laws against stealthing and instead implement conditional consent, as noted in the case of Assange [7]. Regardless, as noted by Ebrahim, stealthing raises questions around what sex is and how it changes with or without a condom, and thus highlighting the need for further research.

The results of this study should be interpreted in the context of its limitations. The college campus is primarily a first-generation minority-serving location, thus the results are not generalizable to other populations. Likewise, to prevent accidental identification of participants, we did not analyze data separately by major. It is highly likely that the results would vary among health science or related majors as compared to non-science-based majors, as the former may be exposed to more sexual health classes. Furthermore, self-reported data is susceptible to social desirability, especially due to the sensitive nature of the topic.

## 5. Conclusions

The results of our study are the first to provide empirical evidence on the current knowledge and perception of stealthing among college students. We note that stealthing, despite being popular among the young adult population, is viewed negatively and raises questions among the population, including concerns over trust, personal health, as well as consent. These results provide evidence that stealthing—the act of nonconsensual condom removal—must become an imperative component of public health efforts to ensure appropriate sexual health literacy. Criminalizing stealthing and making it an equivalent of rape, as noted by many participants in our study, is not a feasible first step and opponents may argue that the consent for sexual intercourse supersedes that of condom removal. However, our results do demonstrate that young adults perceive the act of stealthing as at least a sexual violation—a notion that has been brought up in the field of law and thus provides public health a foundation upon which to initiate health education programs to address stealthing prevention behaviors as well as further research. Specifically, formalized social media qualitative analyses would be beneficial to highlight emergent themes related to how stealthing is discussed, as well as research among stealthers into the driving factors behind such behavior that may shed further light on means of preventive measures. Given the substantial media coverage on stealthing and the importance of ensuring that young adults receive appropriate evidence-based information, public health efforts are needed to incorporate stealthing as a part of sexual health education.

## Figures and Tables

**Table 1 ijerph-17-03527-t001:** Identified themes and associated domains from focus group results.

Domain	Themes	Additional Example Quotes
Knowledge	Awareness	*“…I don’t really know how to describe it. It just sounds negative”.*
		*“…it’s like they are tricking you into something.”*
	Health-decision making	*“..I mean condoms are essentially used for prevention, you’re removing that barrier, you’re exposing yourself to a bunch of diseases now”.*
		*“…when you wear a condom it generally means you guys are protecting each other from any diseases, once you take off the condom you are no longer protected.”*
		*“Yeah, I feel having a condom present in the first place, sets a mutual understanding that we are trying to avoid getting any disease, so if you are willing to take it off, then essentially, you are prompting it.”*
Consent	Communication	*“…It goes back to communication, communication is key and not talking about it and telling them what you are doing or why you did it make me not want to do that again with that person because of trust.”*
		“*…because if I choose to have sex with someone, I am obviously going to want to use a condom for my safety and health because not everyone is up front and honest with their past or if they’ve been tested before, and not everyone is willing to share that information with the other person.”*
	Violation/assault	“*Things such as rape is against the law, I feel this goes hand in hand with stealthing because you removed the condom without their consent, without their knowledge”*.
		*“If you end up with HIV you’re stuck with it for the rest of your life and if you contracted the disease through an act like stealthing I would consider it a crime because you are hurting the other person’s health either way.”*
Social norm	Porn	*“I feel that the viewer will see stealthing in porn and think, oh it’s seems okay because the actor is doing it, then when I practice sex, it’ll be okay to do the same.”*
		*“A lot of people will follow what others do or say, whoever watches porn and see that a condom was being used then removed without the consent of their partner then they will think it is okay to do the same.”*

**Table 2 ijerph-17-03527-t002:** Results from quantitative survey on understanding consent.

**Unwanted touching, kissing, or hugging should be considered sexual assault**	**Total**	**Male**	**Female**
Strongly disagree/disagree	7.9%	10.6%	3.6%
Strongly agree/agree	91.1%	89.5%	96.4%
**Removal of condom without approval of sexual partner should be considered sexual assault**			
Strongly disagree/disagree	16.4%	17.9%	14.5%
Strongly agree/agree	83.6%	82.1%	85.4%

**Table 3 ijerph-17-03527-t003:** Results from quantitative survey on self-efficacy of consent.

Questions	Total	Male	Female
**I am confident in my ability to ask for consent during sexual activity**			
Strongly disagree/disagree	4%	4.3%	3.6%
Strongly agree/agree	96%	95.7%	96.4%
**I am confident in my ability to give consent during sexual activity**			
Strongly disagree/disagree	5.3%	5.3%	5.4%
Strongly agree/agree	94.7%	94.8%	94.5%
**I am confident in my ability to say no during sexual activity**			
Strongly disagree/disagree	3.3%	3.2%	3.6%
Strongly agree/agree	96.7%	96.9%	96.3%

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
