# Peer review of "“You Do It without Their Knowledge.” Assessing Knowledge and Perception of Stealthing among College Students"

_ijerph, 2020, doi:10.3390/ijerph17103527_

Round 1

Reviewer 1 Report

This article tackles an important issue we currently face in responding to the crisis of sexual violence. Where ‘stealthing’ sits along the continuum of sexual violence is the subject of international debate and discussion. In this way, the topic of the paper does address an important and timely issue. However, this paper is not suitable for publication.

Importantly, the paper does not contextualise the discussion in the existing debates around stealthing, and as a result misstates the current state of knowledge on the phenomena. For example, the methods section notes that sexual consent emerged as a central discussion point of the focus groups, but the considerable academic debate as to whether stealthing is in fact rape because it fundamentally negates any consent that was given to sex with a condom, is barely explored.

As such, the article does not demonstrate that the authors are across the issues and literature exploring the phenomena of stealthing, and thus cannot provide any real insight into understandings of the problem. The same can be said for other issues that are flagged and not interrogated throughout the article. For example the paper mentions pornography and says ‘which can be socially normal for young people’. This gives us little insight.

The article seems to argue that lack of knowledge of the term stealthing has a direct link to their knowledge of the practice – but this is not correct. The conclusion is also drawn that this supposed lack of knowledge, coupled with the group being largely immigrants, means that ‘targeted health education to address sexual health literacy, especially condom use and consent may be imperative’. It is unclear why this conclusion was drawn since the group seemed overtly aware of the risk of STI transmission, and this was arguably the most dominant theme that emerged in the focus groups.

The title poses the question ‘is stealthing the next public health emergency?’, but there is no real attempt to answer this question. Instead, the article draws conclusions about criminalisation. The claim is made that ‘These results provide the first evidence that stealthing, the act of nonconsensual condom removal, must become an imperative component of public health efforts in reducing STIs and promoting healthy sexual behavior.’ – I don’t think this has been argued, and it is certainly not the first evidence that could be drawn on to make that argument.

Some other points:

  • Seems to construct stealthing as a social norm – of which it is not, and there is no argument made for this in the article, nor identifiable way the data can make that claim.
  • Refers to stealthing as ‘popular’ – how do we know this (and what does it mean)?
  • Description of the use of the term stealthing as ‘novel’ is inappropriate
  • ‘We considered lack of new codes as reaching thematic saturation, and thus concluded the focus groups’ – It is unclear how these are related.
  • Is half of the table missing?
  • The discussion of findings needs to tell us why the findings are important. For example, the paper says, ‘Among the 153 participants in the survey, 5.2% reported stealthing was removal of condom without consent, while 8.5% associated with the word with a negative connotation, with no significant sex differences.’ What did the rest of the participants say? What does this say about knowledge?
  • Proofreading required. For example, in the conclusion is says ‘However, the lack of actually understanding consent is evident from the aforementioned assessment of unwanted touching, kissing, etc.’ but this was actually mentioned after this sentence. There are also numerous typos and errors.
  • Some conclusions appear drawn from the subjective understandings of the researchers and not from an interrogation of the literature. For example, at one point the claim is made that ‘If individuals have the inability to say no to a sexual activity, then the lack of confidence in giving and asking consent should be lot lower than what the results displayed; thus, further demonstrating a disconnect among college student population on the concept of consent versus saying no.’ I think the college student population are correct on this one – ‘saying no’ and giving and asking for consent are distinct things.

Author Response

Reviewer 1

This article tackles an important issue we currently face in responding to the crisis of sexual violence. Where ‘stealthing’ sits along the continuum of sexual violence is the subject of international debate and discussion. In this way, the topic of the paper does address an important and timely issue. However, this paper is not suitable for publication.

Importantly, the paper does not contextualise the discussion in the existing debates around stealthing, and as a result misstates the current state of knowledge on the phenomena. For example, the methods section notes that sexual consent emerged as a central discussion point of the focus groups, but the considerable academic debate as to whether stealthing is in fact rape because it fundamentally negates any consent that was given to sex with a condom, is barely explored.

Response: Thank you for your feedback. While there remains legal and academic debate on what is consent, per grounded theory, we do have to report on emergent themes, and we cannot delete it regardless of the existing debate. However, we have updated our paper to address this as well as literature that notes where academic research is needed on this.

As such, the article does not demonstrate that the authors are across the issues and literature exploring the phenomena of stealthing, and thus cannot provide any real insight into understandings of the problem. The same can be said for other issues that are flagged and not interrogated throughout the article. For example the paper mentions pornography and says ‘which can be socially normal for young people’. This gives us little insight.

Response: Thank you for your feedback. The goal of this paper was not to conduct a scoping review or systematic review, but to report the results of a mixed method study on the current knowledge and perception. We would like your feedback on what further insight would help regarding pornography, as we reported emergent themes from participants. Per reviewer 3 feedback, however, we did address this in additional details in the discussion.

The article seems to argue that lack of knowledge of the term stealthing has a direct link to their knowledge of the practice – but this is not correct. The conclusion is also drawn that this supposed lack of knowledge, coupled with the group being largely immigrants, means that ‘targeted health education to address sexual health literacy, especially condom use and consent may be imperative’. It is unclear why this conclusion was drawn since the group seemed overtly aware of the risk of STI transmission, and this was arguably the most dominant theme that emerged in the focus groups.

Response: Thank you for your feedback. We did not throughout the paper mention a direct link between knowledge and practice as our study was not approved by institutional review board to assess practice due to the sensitive nature of the topic. As noted in the results, the group was not well aware of STI, as we did not ask questions on STI transmission, but the participants did note, and thus an emergent theme was, the risk of STI due to lack of condom use.

The title poses the question ‘is stealthing the next public health emergency?’, but there is no real attempt to answer this question. Instead, the article draws conclusions about criminalisation. The claim is made that ‘These results provide the first evidence that stealthing, the act of nonconsensual condom removal, must become an imperative component of public health efforts in reducing STIs and promoting healthy sexual behavior.’ – I don’t think this has been argued, and it is certainly not the first evidence that could be drawn on to make that argument.

Response: Thank you for your feedback. The emergent theme of criminalization came from participants and thus were noted, as was reference given, and additional added, to international laws currently being updated to address criminalization; we felt this reference was needed to provide context to what participants have reported.

Some other points:

  • Seems to construct stealthing as a social norm – of which it is not, and there is no argument made for this in the article, nor identifiable way the data can make that claim.
  • Response: We have added additional references in the introduction to demonstrate the popularity of the behavior.
  • Refers to stealthing as ‘popular’ – how do we know this (and what does it mean)?
  • Response: We have added additional references in the introduction to demonstrate the popularity of the behavior.
  • Description of the use of the term stealthing as ‘novel’ is inappropriate
  • Response: We noted in our introduction that the use of the term stealthing is novel but practice is not and we have updated the section with additional context and reference.
  • We considered lack of new codes as reaching thematic saturation, and thus concluded the focus groups’ – It is unclear how these are related.
  • Response: We have clarified how saturation is reached.
  • Is half of the table missing?
  • Response: This maybe in reference to the additional empty line in table 1. We have formatted it correctly in this revision.
  • The discussion of findings needs to tell us why the findings are important. For example, the paper says, ‘Among the 153 participants in the survey, 5.2% reported stealthing was removal of condom without consent, while 8.5% associated with the word with a negative connotation, with no significant sex differences.’ What did the rest of the participants say? What does this say about knowledge?
  • Response: Since 5.2% reported correctly, the rest is incorrect. The same applies to all and we have added a clarifying statement in the results.
  • Proofreading required. For example, in the conclusion is says ‘However, the lack of actually understanding consent is evident from the aforementioned assessment of unwanted touching, kissing, etc.’ but this was actually mentionedafter this sentence. There are also numerous typos and errors.
  • Response: We have re-read and updated the manuscript.
  • Some conclusions appear drawn from the subjective understandings of the researchers and not from an interrogation of the literature. For example, at one point the claim is made that ‘If individuals have the inability to say no to a sexual activity, then the lack of confidence in giving and asking consent should be lot lower than what the results displayed; thus, further demonstrating a disconnect among college student population on the concept of consent versus saying no.’ I think the college student population are correct on this one – ‘saying no’ and giving and asking for consent are distinct things.
  • Response: We have clarified this section further. Participants were asked if they felt confident in saying no. A higher percent reported disagree/strongly disagree, which did not match with asking and giving consent. While we do not say “saying no” is the same as consent, saying no, is, however, part of consent definition. We have provided reference to address this and clarified our analysis.

Reviewer 2 Report

My overall impression is that this is a useful, potentially important paper for its focus on a theme of some novelty (stealthing). That novelty compensates somewhat for the fact that this is a small empirical study (13 focus group participants and 153 survey respondents) that offers new information about the extent to which students at a university are aware (or not) of the practice of removing a condom without one’s partner’s knowledge/consent.

The piece shines an interesting light on the fact that the appropriate (or actual) status of this practice vis-à-vis sexual offences is still something of an unknown, and a nationwide view seems unlikely given the variations between different state laws governing the meaning of sexual consent.

I would say though that changes and improvements are necessary before it can be recommended for publication. Below I offer the following suggestions for strengthening the piece for publication:

  1. The paper moots, but doesn’t really establish or substantiate, that porn impacts on young people’s perceptions of stealthing. I wasn’t so sure that the few remarks on the subject returned in the data justify the accusation in this case (and porn is in any case such an easy target for attacks of this nature), especially given the high number of respondents who had only the vaguest notion of what the practice was. Porn more often than not depicts unprotected sex, but that doesn’t necessarily have anything to do with stealthing, or that it is (by intention or effect or both) ‘glamorizing’ the deceptive nature of the practice.

  1. Case-law in England has developed a line of authority on when failing to use a condom contrary to the wishes or instructions of a sexual partner might vitiate consent. In light of the references in the paper to international developments, I would suggest that the judgments of Assange [2011] EWHC 2849 (Admin) (deception as to condom use) and Re (F) v Director of Public Prosecutions [2013] EWHC 945 (Admin) (deception as to withdrawal before ejaculation) are worth a look in further strengthening these considerations in the paper.

  1. There is a wider literature on young people’s perceptions and understanding of sexual consent (as well as on the impact of porn on sexual attitudes and behaviours). I would suggest that the breadth of literature referred to could be wider to better illustrate the points of contact between this research and other previous work, and to substantiate some of the points made.

  1. The paper suggests, but again doesn’t really take on in any detailed way, that ‘unwanted’ behaviours such as hugging may constitute ‘sexual assault’. What wasn’t so clear to me is what the relevance and implications of such an observation are for this piece? How does ‘unwanted’ tally with ‘un-consented-to’, conceptually? This point could be linked to the previous one (above) about breadth of secondary literature consulted. Suggest taking a look at, for example, Peterson and Muehlenhard, ‘Conceptualizing the ‘wantedness’ of women's consensual and nonconsensual sexual experiences: Implications for how women label their experiences with rape’ (2007) 44(1) Journal Of Sex Research 72

  1. I thought the language of ‘next public health emergency’ to be overblown and wholly inappropriate, given the lack of information on just how prevalent the practice really is, and that we are just now probably entering into a genuine international public health emergency in the form of Covid-19. I understand the authors want to emphasise the importance of their chosen topic, and they are right to seek to do that. However I see no basis or justification for this language of ‘emergency’. (Even if it does turn out to be an emergency, this paper does not have the data to answer the question posed in its own title in any case).

Author Response

Reviewer 2

My overall impression is that this is a useful, potentially important paper for its focus on a theme of some novelty (stealthing). That novelty compensates somewhat for the fact that this is a small empirical study (13 focus group participants and 153 survey respondents) that offers new information about the extent to which students at a university are aware (or not) of the practice of removing a condom without one’s partner’s knowledge/consent.

The piece shines an interesting light on the fact that the appropriate (or actual) status of this practice vis-à-vis sexual offences is still something of an unknown, and a nationwide view seems unlikely given the variations between different state laws governing the meaning of sexual consent.

I would say though that changes and improvements are necessary before it can be recommended for publication. Below I offer the following suggestions for strengthening the piece for publication:

 Response: Thank you, we have noted the updates below.

  1. The paper moots, but doesn’t really establish or substantiate, that porn impacts on young people’s perceptions of stealthing. I wasn’t so sure that the few remarks on the subject returned in the data justify the accusation in this case (and porn is in any case such an easy target for attacks of this nature), especially given the high number of respondents who had only the vaguest notion of what the practice was. Porn more often than not depicts unprotected sex, but that doesn’t necessarily have anything to do with stealthing, or that it is (by intention or effect or both) ‘glamorizing’ the deceptive nature of the practice.

 Response: Thank you for the feedback. The issue of porn was raised and provided qualitatively for participants and we only reported quotes from participants without noting or making statements against porn industry. However, per reviewer 3 comments, we did address it in the future suggested research. The reason for addressing porn, which has been further clarified in the introduction section, is that porn has often included stealthing, but viewers may lack the knowledge or understanding (especially at young age) that it is still actors giving consent. We have clarified this further in our discussion.

  1. Case-law in England has developed a line of authority on when failing to use a condom contrary to the wishes or instructions of a sexual partner might vitiate consent. In light of the references in the paper to international developments, I would suggest that the judgments of Assange [2011] EWHC 2849 (Admin) (deception as to condom use) and Re (F) v Director of Public Prosecutions [2013] EWHC 945 (Admin) (deception as to withdrawal before ejaculation) are worth a look in further strengthening these considerations in the paper.

 Response: Thank you for the feedback. We have added reference and discussion of the paper in the introduction section.

  1. There is a wider literature on young people’s perceptions and understanding of sexual consent (as well as on the impact of porn on sexual attitudes and behaviours). I would suggest that the breadth of literature referred to could be wider to better illustrate the points of contact between this research and other previous work, and to substantiate some of the points made.

  Response: Thank you for the feedback. We have updated manuscript accordingly by taking into account all comments by all reviewers.

  1. The paper suggests, but again doesn’t really take on in any detailed way, that ‘unwanted’ behaviours such as hugging may constitute ‘sexual assault’. What wasn’t so clear to me is what the relevance and implications of such an observation are for this piece? How does ‘unwanted’ tally with ‘un-consented-to’, conceptually? This point could be linked to the previous one (above) about breadth of secondary literature consulted. Suggest taking a look at, for example, Peterson and Muehlenhard, ‘Conceptualizing the ‘wantedness’ of women's consensual and nonconsensual sexual experiences: Implications for how women label their experiences with rape’ (2007) 44(1) Journal Of Sex Research 72

  Response: Thank you for the feedback. We have added why unwanted behaviors were added and added clarifying statements along with reference.

  1. I thought the language of ‘next public health emergency’ to be overblown and wholly inappropriate, given the lack of information on just how prevalent the practice really is, and that we are just now probably entering into a genuine international public health emergency in the form of Covid-19. I understand the authors want to emphasise the importance of their chosen topic, and they are right to seek to do that. However I see no basis or justification for this language of ‘emergency’. (Even if it does turn out to be an emergency, this paper does not have the data to answer the question posed in its own title in any case).

 Response: Thank you for the feedback. When the manuscript was written there was no indication of a pandemic nor could we predict it. However, there was substantial discussion of the #metoo movement as sexual consent and as such, it was appropriate to feel it could be the next issue. However, given the issue of a pandemic, we have updated the title to reflect more appropriate title.

Reviewer 3 Report

Review of the article: “You do it without their knowledge.” Is stealthing the next public health emergency?

I consider it an article on a very relevant topic and it is very well written. Consent for sexual relations is currently a very important issue to be approached from several perspectives and fields, such as the medical, the social and even the legal field, which the article also refers to.

The social media perspective has an important point into the analysis. The influence of peer groups is an important aspect, as research already shown the importance of popularity for individual and collective behaviors. That is why I consider it important to bring this issue to public attention. The effect of the public debate and group pressure has an effect, both on the creation of a specific social behavior, and on enhancing the rape culture.

The article is very well done, at the structure level it is perfectly written. The introduction is well explained, although perhaps it is a bit short. I think, concepts such as “bragging”, “reddit” or “spreading seeds” are relevant to show how men encourage each other and social networks promote their actions. I think this point of popularity in the media is very relevant in the analysis of achieving consent in sexual relations. One of the issues to be raised here is how attractive it gets to be, the fact of getting popular, to be cool the fact of doing “stealthing”. In this regards, when authors say (page 2) “the act is morally wrong”, I wonder how attractiveness and sexual-affective attractiveness comes to the debate. From the ethical and moral perspective, the analysis is clear, however, from the attractiveness perspective, how do we change it and make socially attractive an alternative kind of having sex? (perhaps this is just a reflection, to think about more that than to address here)

Sexual harassment and sexually transmitted infections are also important points raised in the article. Indeed, it is clear that this article discuss a tremendously relevant topic. To improve the article, I recommend including a section of State of the art, or Theoretical Framework, between introduction and the Methods section. However, authors have strictly followed the “research manuscript sections” (introduction, methods, results, discussion, conclusions).

The methods section is also very well done and well explained. The mixed methods approach is indeed relevant in this case. The interviews, focus groups, the questions addressed to the people are properly included. Even some ethical issues such as consent forms are included. To improve this section of the article, regarding data analysis, when it says: “modified grounded theory” it would be great to quote someone who supports this methodology/theory, or to quote a project, article, … in which it has been used before. It cannot be taken for gradated that people/readers know what the article is talking about, or, in this case, how information has been analyzed.  

Results are very clear and very well presented. I think the topics chosen to expose the results (awareness, health-decision making, consent, social norm) are right and corrected. There is a social issue, with sexual implications, however there are healthy risks which need to be approached too, and they are. The quotes are right and clear, which make contribution to be relevant, I really like how clear the quotes are and how well they express the term to be explained to the readers. Self-efficacy item (on page 5) does not appear on Table 1 among the items presented there. I understand that this is due to its quantitative approach and results presented there, however it would be great to have a table also for these quantitative results. Regarding table 1, within the section “social norm”, lack of awareness category, does not have any quote. (maybe it should not appear on the table). Data at a quantitative level is poorly described and not so well presented. Results on page 6 (5.2% and 8.5%) are important. The same happens with other items (agree/disagree and the corresponding percentages). They will be clearer with a table rather than just narrated.

The discussion part is also very good. It proposes correctly the mentioned points and the connection between them, from the lack of knowledge to the health problems that stealthing may generate. On a matter of sections and paragraphs, I understand that “Discussion and Conclusions” are presented within the same paragraphs, right? I will suggest authors to divide both sections and create a “Conclusion” section.

Authors can also suggest some contributions or further topics to be researched or addressed, such as, consent training, sexual harassment awareness, “anything less than yes is no”, as well as legal implications of stealthing. In my view, relating consent with sexual assault and stealthing, are relevant issues, also to be approached in terms of legality and legal consequences, in the context of higher education.

Emphasizing all this points, this debate, research and this paper will contribute to change the “social norm”, as authors mention. I really thing this article will contribute to create more awareness on the issue and enable further research.

Author Response

Reviewer 3

I consider it an article on a very relevant topic and it is very well written. Consent for sexual relations is currently a very important issue to be approached from several perspectives and fields, such as the medical, the social and even the legal field, which the article also refers to.The social media perspective has an important point into the analysis. The influence of peer groups is an important aspect, as research already shown the importance of popularity for individual and collective behaviors. That is why I consider it important to bring this issue to public attention. The effect of the public debate and group pressure has an effect, both on the creation of a specific social behavior, and on enhancing the rape culture.

Response: Thank you for your feedback.

The article is very well done, at the structure level it is perfectly written. The introduction is well explained, although perhaps it is a bit short. I think, concepts such as “bragging”, “reddit” or “spreading seeds” are relevant to show how men encourage each other and social networks promote their actions. I think this point of popularity in the media is very relevant in the analysis of achieving consent in sexual relations. One of the issues to be raised here is how attractive it gets to be, the fact of getting popular, to be cool the fact of doing “stealthing”. In this regards, when authors say (page 2) “the act is morally wrong”, I wonder how attractiveness and sexual-affective attractiveness comes to the debate. From the ethical and moral perspective, the analysis is clear, however, from the attractiveness perspective, how do we change it and make socially attractive an alternative kind of having sex? (perhaps this is just a reflection, to think about more that than to address here)

Response: Thank you for your feedback and insight.

Sexual harassment and sexually transmitted infections are also important points raised in the article. Indeed, it is clear that this article discuss a tremendously relevant topic. To improve the article, I recommend including a section of State of the art, or Theoretical Framework, between introduction and the Methods section. However, authors have strictly followed the “research manuscript sections” (introduction, methods, results, discussion, conclusions).

Response: Thank you for your feedback. We acknowledge the importance of the formatting recommendation and we have updated as such.

The methods section is also very well done and well explained. The mixed methods approach is indeed relevant in this case. The interviews, focus groups, the questions addressed to the people are properly included. Even some ethical issues such as consent forms are included. To improve this section of the article, regarding data analysis, when it says: “modified grounded theory” it would be great to quote someone who supports this methodology/theory, or to quote a project, article, … in which it has been used before. It cannot be taken for gradated that people/readers know what the article is talking about, or, in this case, how information has been analyzed.  

Response: Thank you, we have explained how the grounded theory was modified to address emergent themes, as well as added reference and details on how the more modified GT was used.

Results are very clear and very well presented. I think the topics chosen to expose the results (awareness, health-decision making, consent, social norm) are right and corrected. There is a social issue, with sexual implications, however there are healthy risks which need to be approached too, and they are. The quotes are right and clear, which make contribution to be relevant, I really like how clear the quotes are and how well they express the term to be explained to the readers. Self-efficacy item (on page 5) does not appear on Table 1 among the items presented there. I understand that this is due to its quantitative approach and results presented there, however it would be great to have a table also for these quantitative results. Regarding table 1, within the section “social norm”, lack of awareness category, does not have any quote. (maybe it should not appear on the table). Data at a quantitative level is poorly described and not so well presented. Results on page 6 (5.2% and 8.5%) are important. The same happens with other items (agree/disagree and the corresponding percentages). They will be clearer with a table rather than just narrated.

Response: Thank you, we agree the table would make it clear and as such we have added that.

The discussion part is also very good. It proposes correctly the mentioned points and the connection between them, from the lack of knowledge to the health problems that stealthing may generate. On a matter of sections and paragraphs, I understand that “Discussion and Conclusions” are presented within the same paragraphs, right? I will suggest authors to divide both sections and create a “Conclusion” section.

Response: Thank you, we have updated it to include Conclusion section separately.

Authors can also suggest some contributions or further topics to be researched or addressed, such as, consent training, sexual harassment awareness, “anything less than yes is no”, as well as legal implications of stealthing. In my view, relating consent with sexual assault and stealthing, are relevant issues, also to be approached in terms of legality and legal consequences, in the context of higher education.

Response: Thank you for the suggestions. We have added a futured suggested research in the discussion section.

Emphasizing all this points, this debate, research and this paper will contribute to change the “social norm”, as authors mention. I really thing this article will contribute to create more awareness on the issue and enable further research.

Response: Thank you for your feedback, it is well appreciated.

Round 2

Reviewer 2 Report

This revision responds positively and coherently to the comments I made the first time around. The claims made on the strength of the data are more appropriately framed, and certain areas of discussion (particularly around porn) are stronger here.

Just a couple of minor things to correct before finalising: 1) I did spot a few typos which you should iron out; 2) It was a good idea to cite Assange case, but please note this did not create a new legal principle (which the authors call 'conditional consent') -- merely it affirms that under the relevant legislation, consent is (and always was) something that can be understood in conditional terms (the wearing of a condom being the obvious example of a relevant condition, but may include other things, such as not coming inside the vagina during unprotected sex); 3) one particular sentence (near to the end, shortly before the conclusion) struck me as odd and difficult to understand - not sure if this is just a mistake, but it should be checked for syntax:

'For example, Blanco (25) notes that removal of condom without consent removes bodily autonomy of women, however, considering it rape may become over-criminalization such that pleasurable experiences, likely such as intercourse without condom, does not become negated.'

Assuming those few corrections will be made, my view is that this is now fit for publication.

Author Response

This revision responds positively and coherently to the comments I made the first time around. The claims made on the strength of the data are more appropriately framed, and certain areas of discussion (particularly around porn) are stronger here.

Just a couple of minor things to correct before finalising: 1) I did spot a few typos which you should iron out;

Response: We have gone through and re-read the manuscript and updated any issues we have missed previously. Thank you.

2) It was a good idea to cite Assange case, but please note this did not create a new legal principle (which the authors call 'conditional consent') -- merely it affirms that under the relevant legislation, consent is (and always was) something that can be understood in conditional terms (the wearing of a condom being the obvious example of a relevant condition, but may include other things, such as not coming inside the vagina during unprotected sex);

Response: We have update this section and noted the outcomes along with citations.

3) one particular sentence (near to the end, shortly before the conclusion) struck me as odd and difficult to understand - not sure if this is just a mistake, but it should be checked for syntax:

'For example, Blanco (25) notes that removal of condom without consent removes bodily autonomy of women, however, considering it rape may become over-criminalization such that pleasurable experiences, likely such as intercourse without condom, does not become negated.'

Response: We have gone ahead and re-read this section and updated it accordingly.

Assuming those few corrections will be made, my view is that this is now fit for publication.

Response: Thank you for your feedback.